# Comparative Numerical Studies on the Structural Behavior of Buried Pipes Subjected to Extreme Environmental Actions

**DOI:** 10.3390/ma15093385

**Published:** 2022-05-09

**Authors:** Ana Diana Ancaș, Florin-Emilian Țurcanu, Marina Verdeș, Sebastian Valeriu Hudisteanu, Nelu-Cristian Cherecheș, Cătălin-George Popovici, Mihai Profire

**Affiliations:** Building Services Department, Faculty of Civil Engineering and Building Services, Gheorghe Asachi Technical University, 700050 Jassy, Romania; ancas05@yahoo.com (A.D.A.); verdesmarina2003@yahoo.com (M.V.); catalin-george.popovici@academic.tuiasi.ro (C.-G.P.); profiremihai@yahoo.com (M.P.)

**Keywords:** extreme environments, performance evaluation, numerical simulation, structural response

## Abstract

Globally, there are several critical infrastructure networks (water and gas networks) whose disruption or destruction would significantly affect the maintenance of vital societal functions, such as the health, safety, security, and social or economic well-being of people. They would also have significant local, regional, and national impacts as a result of the inability to maintain those functions, and would have similar cross-border effects. The main objective of this article is to investigate by comparative numerical studies the structural response of three types of buried pipes made of different materials, primarily steel, concrete, and high-density polyethylene, resulting from the impact of the environment through exceptional external actions, such as explosions at the surface of the land in the vicinity of the laying areas. The dynamic transient analysis of the equation of motion with the application of the explicit integration procedure was performed with the ANSYS numerical simulation program. This study allows designers to solve complex problems related to the quality of the laying ground of water networks to canals. The knowledge accumulated gives us the possibility to correctly specify the optimal economic and technical value of the ratio between the laying depth of pipes and their diameter, the importance of the radius ratio of the pipe and the thickness of its wall, and, importantly, the improvement of the quality of the foundation ground. Following the results obtained, it is estimated that the optimal economic and technical value of the ratio between the laying depth of the pipes (H) and their diameter (D) is 3, regardless of the material from which the pipe is made.

## 1. Introduction

As the main element of a network (water, sewerage, gas, oil, etc.), a pipeline must have sufficient strength and/or rigidity to perform its intended function in good condition. In the current context, in setting design criteria, an account must be taken of the increasing occurrence of extreme actions. The giant earth movements associated with these actions can be devastating to construction in some areas, including pipelines. Depending on the material these nets are made from, they respond differently. A pipe made of more flexible material with a flexible joint will allow the pipe to conform to the movement of the earth without significant damage. The passage of waves through the ground generates different voltages in the pipe.

Current research shows that only the passage of waves causes damage to rigid pipelines with mechanical joints that are much more sensitive to the transient movement of the earth.

The main objective of the research in this article is to investigate the structural responses of three types of pipes of different materials: steel, concrete, and high-density polyethylene, resulting from the application of an overpressure caused by exceptional gravitational actions.

Fully or partially buried underground structures can be of any shape and destination (shelters, basements, silos, warehouses, tunnel or pipe networks, etc.) and are made of different materials (steel, plain or reinforced concrete, wood, fiberglass, plastics, etc.). For safety reasons, both during operation and during burglary attempts, the location of the pipes is frequently chosen underground. Wavefronts caused by explosions during large-scale armed or terrorist conflicts, accidents caused by the accumulation of explosive gases due to damage or the application of improper operating technologies, and many others, are exceptional actions of great importance commonly neglected in design because they are falsely considered unlikely to occur during the life of the investment. A wavefront with a pressure value of 138 kPa can completely destroy even reinforced concrete structures [1,2]. The explosion of an underground pipeline can have unpredictable consequences, including loss of life. A decommissioning of a water, gas, liquid fuel, or any other supply line can dramatically affect the manufacturing industry (even destroying an entire production line) and the environment through which it passes. Pollution can cause rapid chain damage, and the environment it effects becomes irrecoverable. A Wikileaks report [3] published in the New York Times (USA) revealed that large-scale insurgent and terrorist attacks are imminent around the globe and recommended the development of experimental and numerical research and investigation, calling it imperative for adequate and effective enforcement. All protective or preventive measures may be taken to avoid catastrophes or tragedies.

Since an overpressure caused by the crossing of the pipeline by a weighty vehicle does not cause problems in modeling the gravitational action, the following is a detailed model of the creation of a wavefront with an explosion source. The ground-air-structure interaction is included to simulate a numerical model that is as close as possible to reality.

The data and models were synthesized from the results of tests performed in the field and in prestigious laboratories worldwide that have specialized in such research, as well as from theoretical studies presented in scientific events in recent years in specific fields [4]. The analysis of the current state of research shows that most of the works that focused on the analysis of the quasi-static interaction of land structure, in the hypothesis of infinite, homogeneous, and isotropic half-space, do not fully agree with the results of answers obtained on three-dimensional numerical models required by static actions and seismic. It also revealed that transient dynamic analyses can reveal structural responses much closer to real situations, even assuming homogeneous and isotropic environments with linearly elastic behavior due to high-performance algorithms implemented in computing programs. The exceptionally high complexity of the numerical modeling of all materials, conditions, and interactions required the selection of only those that are relevant and do not cause variations in responses more significant than 5–8%. Experimental analyses have shown that the severity of exceptional actions requires, in practice, the neglect of models or effects that would not improve the response of the structure.

Thus, the numerical research conducted in this study neglected friction and landslide and was limited to determining the ratios between the various values of pressures within the model under the mentioned exceptional actions and to identifying the maximum permissible displacement values of 2%, as exceeding these limits may damage the pipe, which is not permissible. An exploration beyond these limits is not relevant. The analyses considered the geometric nonlinearity effects caused by high amplitude displacements and rotations.

## 2. Theoretical Considerations

### 2.1. Shock Wave Interaction

The shock wave that encounters a solid surface, or an object made of a much denser environment than that of the transported wave, will be reflected by it. Depending on its geometry and size, diffraction around the object will also be recorded. The simplest case is a rigid wall of infinite dimensions hit by a shock wave at an angle of zero incidences [5]. In this case, the frontal wave moving at a speed $U_{s}$ in the air at ambient pressure $P_{0}$ will be reflected by the wall, while the air particles moving forward will enclose the shock wave being returned to rest and then compressed, inducing a reflected overpressure on the wall, which will have a magnitude greater than the value of the incident overpressure (Figure 1).

*U_s_*—wavefront speed at *P*_0_; 

*U_r_*—reflected wave speed;

*p*_0_—ambient air pressure;

*p_r_*—reflected pressure;

*p_s_*—overpressure.

The wavefront parameters were first determined by Rankine and Hugoniot [6] based on the principle of conservation of momentum and energy. With the help of the relations formulated by Rankine and Hugoniot, all the important parameters can be obtained, assuming the air behaves like a real gas, having the ratio of the specific heats $\frac{C_{p}}{C_{v}}=\gamma \;$ [7]. For a zero angle of incidence, the maximum value of the reflected pressure, $p_{r}$, is given by the relation:(1)$$p_{r}=2\;p_{s}+\left(\gamma +1\right)q_{s}$$
where the dynamic pressure $q_{s}$ is:(2)$$q_{s}{}^{2}=\frac{1}{2}p_{s}u_{s}{}^{2}$$

In this relation, $p_{s}$ represents the density of air and $u_{s}$ is the velocity of the particles behind the wavefront. It can be shown that:(3)$$u_{s}=\frac{a_{0}p_{s}}{\gamma p_{0}}{\left[1+\left(\frac{\gamma +1}{2\gamma }\right)\frac{p_{s}}{p_{0}}\right]}^{-\frac{1}{2}}$$
where $a_{0}$ is the speed of sound in the environment. By introducing Equations (2) and (3) in Equation (1) and considering γ=1.4 for the air we obtain the relation:(4)$$p_{r}=2p_{s}\left(\frac{7p_{0}+4p_{s}}{7p_{0}+p_{s}}\right)$$

It is observed that the upper and lower limit values of the reflected pressure can be set $p_{r}$: when the overpressure $p_{s}$ is much lower than the ambient pressure (phenomenon that occurs at a long distance of a low intensity explosion source); thus Equation (4) is reduced to:(5)$$p_{s}\ll p_{0}\;\to p_{r}=2p_{s}$$

If $p_{s}$ is much higher than the ambient pressure (a phenomenon occurring at a short distance from a high-intensity explosion source), then Equation (4) is reduced to:(6)$$p_{r}=8p_{s}$$

If the reflection coefficient, $C_{R}$, is defined as the ratio of the pressures $\;p_{r}$ and $p_{s}$, then the Rankine–Hugoniot relation shows that the value $C_{R}$ is between the limits 2 and 8. However, due to the dissociation effect of the gas molecules, values of up to 20 of the $C_{R}$ coefficient was also measured at very short distances from the explosion source.

### 2.2. Mach Reflection

In previous analyses, it was considered that the explosion wave hits a target structure at a zero-incidence angle. When the angle of incidence is 90°, the reflection phenomenon is not recorded. The surface of the target is loaded with a peak overpressure, sometimes called “lateral” pressure. For the angle of incidence values between these two limits, either an ordinary reflection phenomenon or a Mach-type reflection can occur [5,6]. For a specific deal of the pressure $p_{r}$, there is a limit value of the angle of incidence above which the usual reflections do not occur and the Mach reflection appears. Moreover, for each type of gas, there is a specific value of the angle over which the value $p_{r}$ is greater than the value of the pressure $\;p_{r}\;$ generated by a reflection below a normal angle of incidence (zero). For air, this angle is approximately equal to 40°. For a $p_{s}$ value, there is a certain value of the angle of incidence for which the ratio between $p_{r}$ and $p_{0}$ is minimal. It should be noted that the reflection angle of the wavefront generated by an explosion increases with an increasing angle of incidence.

The Mach Reflection Phenomenon occurs when the angle of incidence exceeds a so-called “threshold value” which depends on the value of the incident overpressure. Mach Reflection is a complex process and is sometimes described as a “wiping” surface effect when the incident wave “slips” over the target surface and does not hit it directly, as in the case of lower values of the incidence angle.

In this reflection process, the reflected wave meets the incident wave at a point above the reflection surface and a third front occurs, forming a tree-like “Mach structure”. The coalescence point of the three wavefronts is called the triple point or Mach point [7]. Behind the “Mach strain” and reflected waves is a region specific to a “tractor wave”. Although the pressure is the same, there are differences between the densities and velocities of the particles relative to the environment. The formation of the “Mach strain” is critical when a conventional or nuclear charge is detonated at a certain height above the ground. Likewise, if an explosive charge is detonated inside a building where the angles of incidence can vary substantially, a “Mach strain” will also form.

It should be noted that at the onset of Mach reflection, there is a gradual increase in overpressure, a process that highlights the importance of forming the “Mach strain”. In the case of Mach reflection for a given overpressure, a much higher level of overpressure is generated than in ordinary reflections [5].

Based on what is presented, the variation of the reflected wavefronts is found, in the case of outdoor explosions, to be depending on the distance from the place of detonation. Thus, at a distance of only 10 m from the site of the explosion, the pressure of the wavefronts decreases significantly so that only measures to protect the objectives by limiting or prohibiting access to a certain area can often be sufficient.

## 3. Materials and Methods

For all these reasons, in this article the main objective is to investigate the structural response for three types of pipes of different materials (steel, concrete and high-density polyethylene—HDPE) obtained from applying an overpressure caused by gravitational operating actions exceptionally (explosions at the surface of the ground in the vicinity of the laying areas of the pipes, etc.). The response of metallic elements to impulse-type actions is strongly influenced, mainly by the duration of the action, but also by other factors, such as structural configuration, type of connection, interactions between subsystems, etc. Research has, because of this, been restricted and focused on transiently dynamic responses with the inclusion of special behavioral properties to such requests [8,9,10].

Numerical research was also carried out in the case of compact, homogeneous, and isotropic soils in the case of a minimum humidity necessary to ensure a minimum degree of compaction of 98% without considering the water pressure in the pores, but by taking into account a nonlinear geometric behavior [11,12,13,14]. In the presence of saturated clays, the presence of water in the pores can significantly influence the response of buried structures—such as pipes—in the case of very short impulse actions, usually tens of seconds or milliseconds caused by explosions or impact forces.

This direction of exploitation and expansion of numerical and experimental research is currently the focus of many researchers around the world so as to be able to identify all the vulnerabilities of water, gas, fuel, etc., transmission networks. Researchers aim to identify these weaknesses in time to apply precautionary and protective measures that are challenged and minimize the risk of accidental or intentional destruction.

Figure 2 presents the notations and hypotheses for modeling underground pipes:

Where:

D, R—diameter and radius of the pipe;

H_sup_—the height of the ground above the pipe;

H_inf_—the height of the ground under the pipe;

T—pipe wall thickness.

Comparative numerical studies have been performed on three types of hollow underground pipes—steel, concrete, high-density polyethylene, and smooth—in the case of applying a gravitational overpressure caused by a wavefront induced by an explosion at the surface of the ground. The dynamic transient analysis of the equation of motion with the application of the explicit integration procedure was performed with the ANSYS calculation program [15,16] on a total of 108 finite element numerical models, as follows:➢A total of 15 numerical models for the analysis of the laying of pipes in loose sand in the hypothesis of varying the depth of burial of the pipe from H/D = 1 to H/D = 5.➢A total of 15 numerical models for the analysis of the laying of pipes in compacted sand in the hypothesis of the variation of the depth of burial of the pipe from H/D = 1 to H/D = 5.➢A total of 15 numerical models for the analysis of the positioning of the pipes in compacted clay in the hypothesis of the variation of the depth of burial of the pipe from H/D = 1 to H/D = 5.➢A total of 21 × 3 = 63 numerical models for the case of underground laying of compacted clay pipes at a ratio H/D = 3, assuming the variation of the ratio between the radius of the pipe and its wall thickness from 3 to 25.

Preliminary numerical studies have shown that the vulnerability of the pipes is much higher if they are empty than in the operating situation, where the internal pressure of the transported gas or liquid favors their behavior in gravitational actions. The three-dimensional models for simulating the numerical response of a 2 m long pipe section were made up of a variable number of finite elements, between 19,200 and 50,400, depending on the H/D ratio.

The gravitational overpressure applied to the top of the model was considered to be 1/3 of the length of the pipe in order to be able to consider the effects of spatial cooperation between the spatial finite elements of type *Solid 281* and those for modeling the plane state of tension type *Shell 28*. The values of the final results were extracted from the middle section of the numerical model where, as was natural, the extreme values were recorded.

Figure 3 shows some answers of the numerical models of a buried pipe under the conditions of gravitational application of a dynamic action (middle section of the numerical model).

For the first 45 numerical models the considered diameter of the pipes was 60 cm, and for the other 63 numerical models the dimensional variation was achieved both by modifying the discretization step and by scaling the model.

The mechanical characteristics of the pipes and the foundation ground were considered according to Table 1 and Table 2.

The constitutive models of the materials used in numerical analysis were introduced in accordance with the results of the latest research obtained in various laboratories around the world and included the property of sensitivity of materials to actions applied at very high speeds.

Figure 4 presents the constitutive model stress-specific deformation of the steel [17].

Since in relation to the other component materials of the numerical model the specific stress–strain ratio of the concrete (Figure 5, [18,19,20]) changes only at much higher rates of application of the loads than those reached by the pipe in concrete to be requested, the constitutive model of the concrete was chosen similar to the quasi-static demands [21,22].

Figure 6 presents the constitutive model of HDPE pipes—structural response to high loading speeds [23,24,25].

Figure 7 presents a generic stress–strain constitutive model specific to all three types of terrain used in the numerical model.

In Figure 8 the most probable response curves were selected [5]. Regarding the ratio between the explosive materials, it is specified that an explosion of 500 kg of ammonium nitrate (a material frequently used in terrorist attacks) is equivalent to detonating 60 kg of TNT to achieve a highly destructive effect. Therefore, it is necessary to transport many explosives, limiting of transport size being one of the most effective protection measures.

## 4. Results and Discussion

The analysis of the numerical model shown in Figure 1 took into account the value of a reflected wavefront pressure of 2500 kPa, at a distance of 10 m from the site of the explosion (with appropriate propagation velocities, depending on the environment in which the pipes were laid), equivalent load—according to the corresponding distribution on the exposed surface—with a gravitational overpressure given by a very large transport vehicle, having the axle load of approximately 100 tons.

The results of the analyses performed are summarized in graphical form:

### 4.1. Variation of Pressures Depending on the Laying Floor of the Pipe and the Type of Material

Figure 9a,b shows the variation of pressures at the top and bottom of the pipes depending on the depth of their laying in loose sand.

Figure 10a–c presents the variation of the pressures at the upper, lower, and middle part of the pipes depending on their laying depth in compacted sand.

Figure 11a–c presents the variation of the pressures at the upper, lower, and middle part of the pipes depending on their laying depth in compacted clay.

### 4.2. Absolute Displacement Variation Depending on the Laying Floor of the Pipe and the Type of Material

Figure 12a–c presents the variation of the absolute displacement at the upper, lower, and middle parts of the pipes depending on their laying depth in loose sand.

Figure 13a–c shows the variation of the absolute displacement at the top, bottom, and middle of the pipes depending on the depth of their laying in compacted sand.

Figure 14a,b shows the variation of the absolute displacement at the top and bottom of the pipes depending on their laying depth in compacted clay.

Based on all the analyses performed and the interpretation of the obtained graphs, it was observed that the quality of the laying pipe is of great importance, showing that a high degree of compaction will ensure better predictability of the response and a structural behavior between the estimated limits.

HDPE pipes have a more predictable behavior than steel or concrete pipes, proportional to the laying depth and quasi-linear after exceeding the ratio H/D = 3. Analyses show that all pipes have a more pronounced specific crushing mode at the top (similar to yielding to static flattening stresses) and lower in the middle areas as the ground pressure balances the thrusts. Therefore, the ratio between the radius of the pipe *R* and the thickness of its wall *t* is of great importance.

In the case of steel pipes, the increase of the *R*/*t* ratio also determines an increase of displacements. In contrast, in concrete and HDPE, there is generally a quasi-stabilization and even a decrease in values in some cross section areas.

Suppose pipes laid in loose sandy soils are identified at sites. In that case, it is recommended either to improve their geotechnical qualities by compaction or by applying injection technologies of various substances or to completely replace these layers, preferably with clays that achieve a degree of compaction of at least 98%.

## 5. Conclusions

According to the obtained graphs, it is estimated that the optimal economic and technical value of the ratio between the laying depth of the pipes (H) and their diameter (D) is 3, regardless of the material from which the pipe is made. Increasing the laying depth of the pipe causes small pressures and displacements in the pipe in the case of impulse-type actions such as those mentioned (explosions).

The ratio between the radius of the pipe *R* and the thickness of its wall *t* is of great importance and it is recommended that it be over 10, both in terms of behavior in operating actions and for reasons of predictability of the response. If the *R*/*t* ratio is below 10, the displacements are difficult to estimate, as it is possible that the phenomena of local veiling or specific deformation of thick-walled tubes alter the response, assessing a desired degree of safety by designing such cases being random.

## Figures and Tables

**Figure 1 materials-15-03385-f001:**
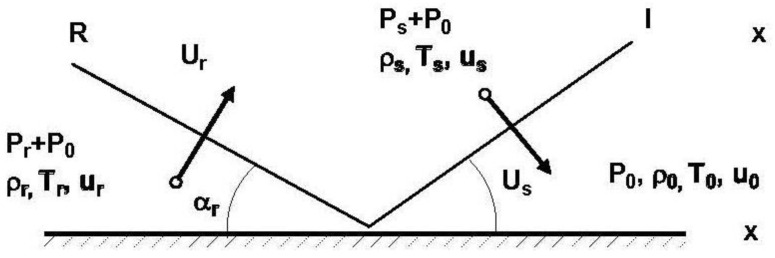
The reflection of a wavefront at the meeting of a rigid and infinitely long environment.

**Figure 2 materials-15-03385-f002:**
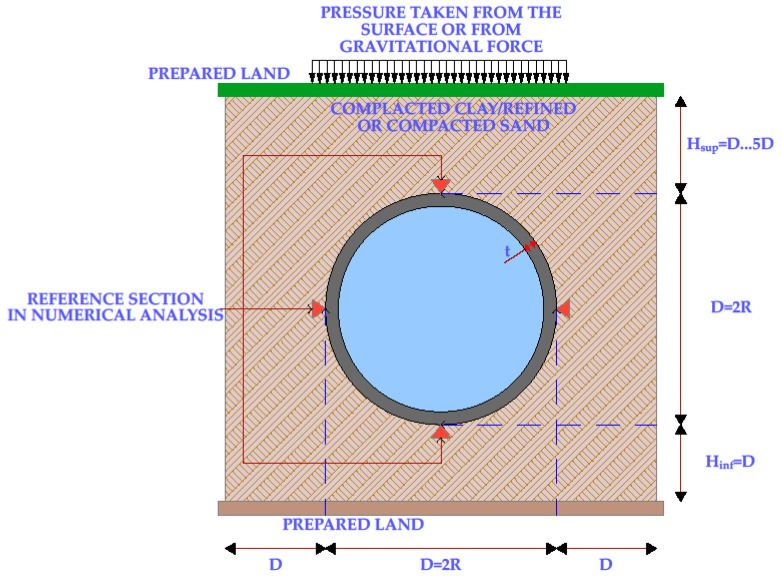
Notations and hypotheses for modeling underground pipes.

**Figure 3 materials-15-03385-f003:**
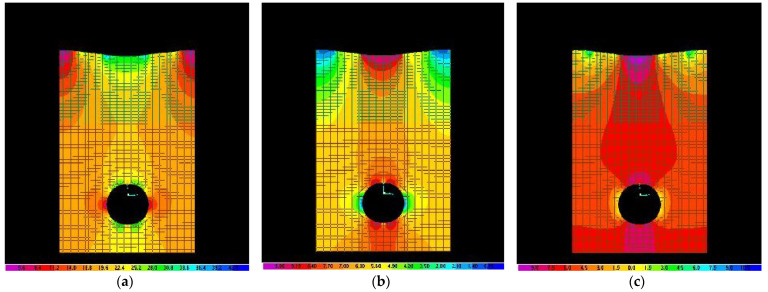
Examples of answers of numerical models representing a buried pipe in the hypothesis of the gravitational application of a dynamic overpressure: (**a**,**b**)—main normal stress (N/mm^2^), (**c**)—absolute displacements (mm).

**Figure 4 materials-15-03385-f004:**
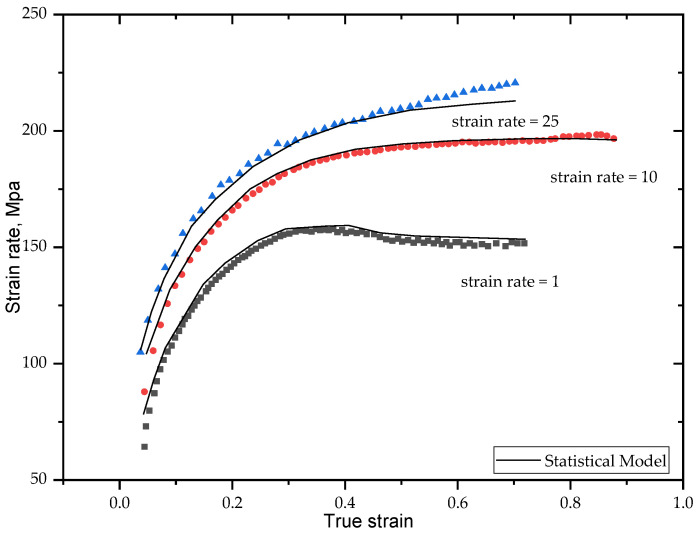
Constitutive model stress—specific deformation of the steel.

**Figure 5 materials-15-03385-f005:**
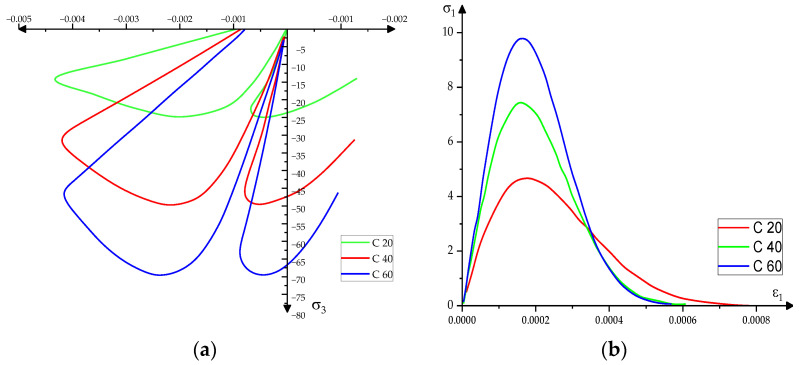
The constitutive model of the concrete required for impulse-type actions depending on the grade of concrete C20, C40, C60 (**a**) tension curves-specific deformation at uniaxial compression (**b**) tension curves-specific deformation at a uniaxial stretch.

**Figure 6 materials-15-03385-f006:**
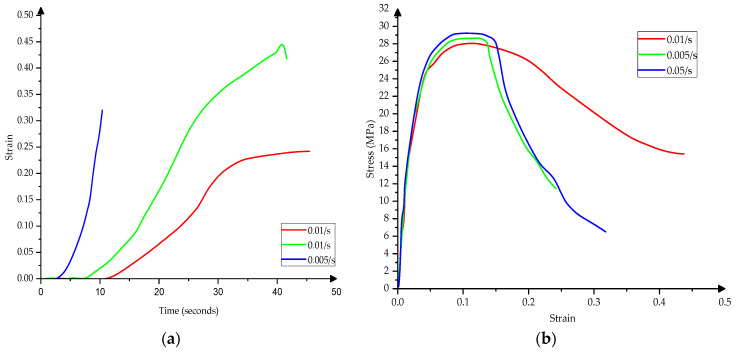
The constitutive model (**a**) Strain–Stress and (**b**) Stress–Strain of HDPE pipes.

**Figure 7 materials-15-03385-f007:**
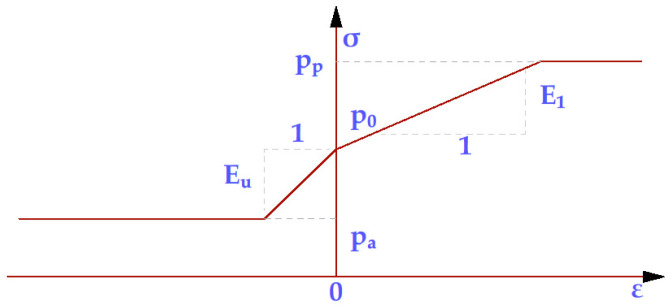
Generic stress–strain constitutive model-specific deformation for all three types of soils used in the numerical model (loose sand, compacted sand, saturated clay).

**Figure 8 materials-15-03385-f008:**
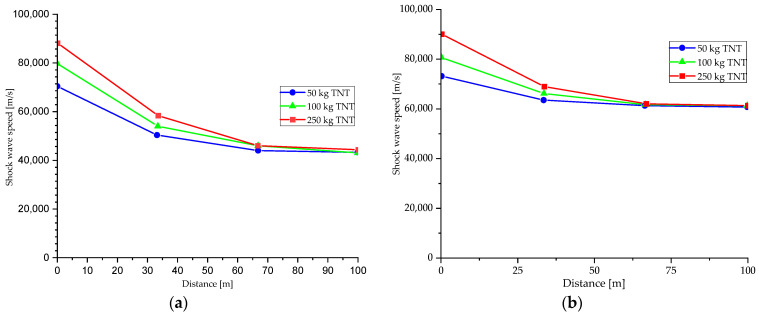
Shock wave propagation speed variation (**a**) in sandy soils (**b**) in clay soils.

**Figure 9 materials-15-03385-f009:**
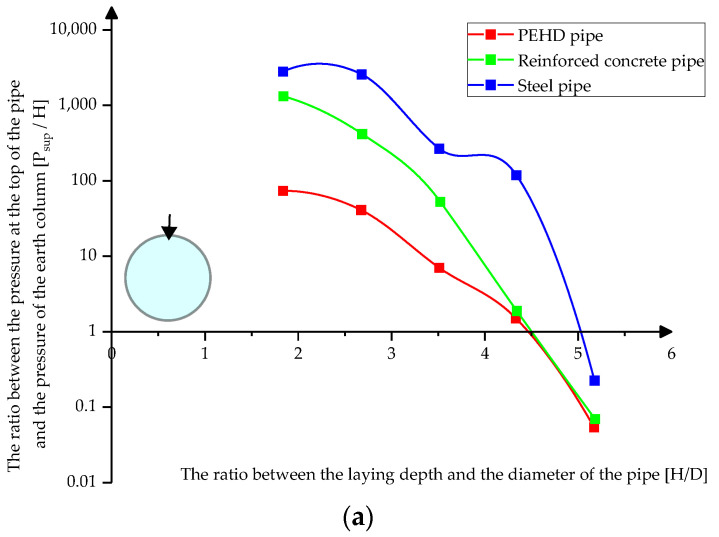
Variation of pressures on pipes in loose sands (**a**) at the top of the pipes (**b**) at the bottom of the pipes.

**Figure 10 materials-15-03385-f010:**
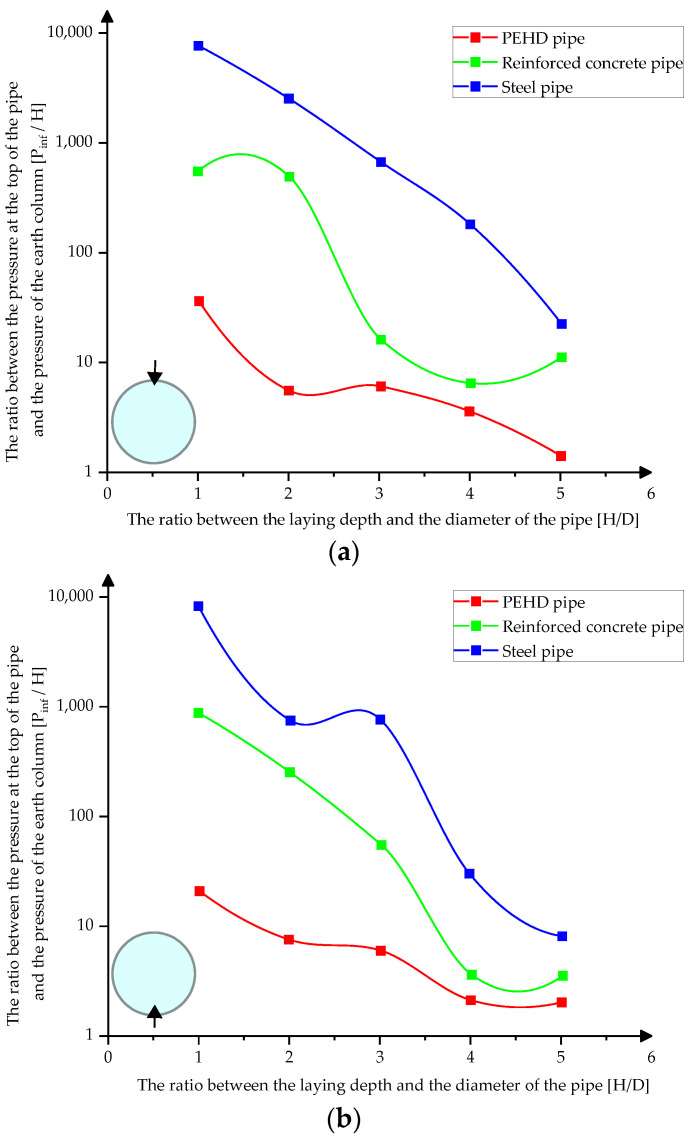
Variation of pressures on pipes in compacted sands (**a**) at the top of the pipes (**b**) at the bottom of the pipes (**c**) in middle sections.

**Figure 11 materials-15-03385-f011:**
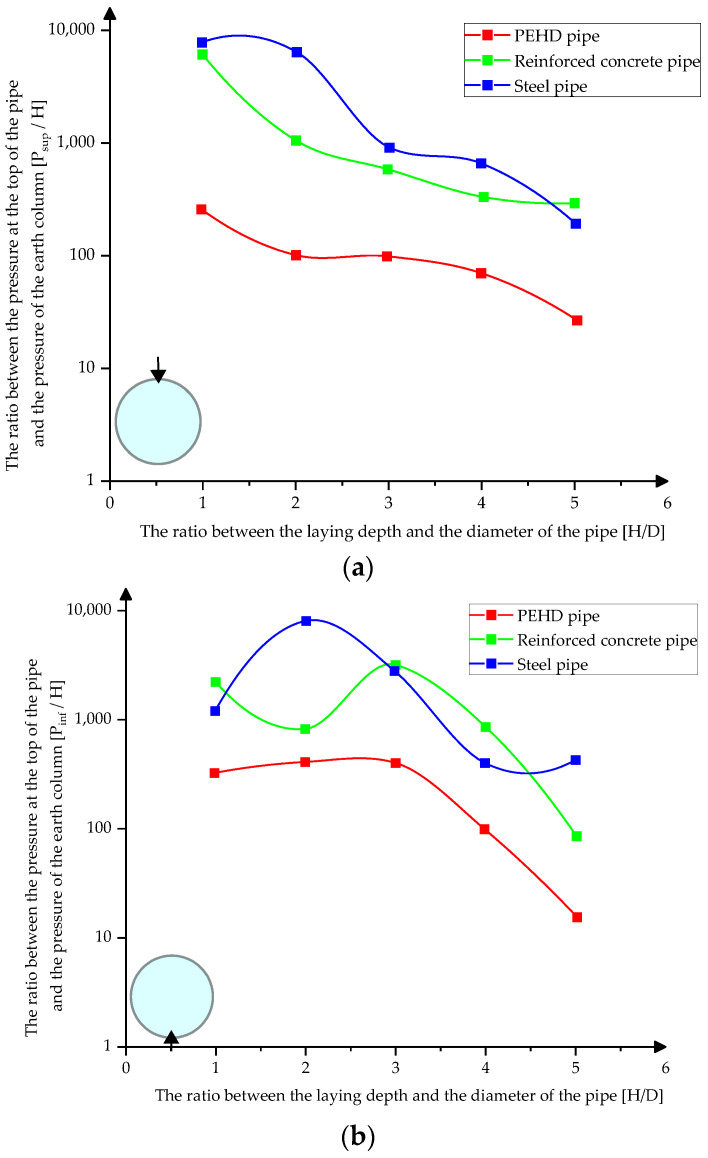
Variation of pressures on pipes in compacted clay (**a**) at the top of the pipes (**b**) at the bottom of the pipes (**c**) in the middle sections.

**Figure 12 materials-15-03385-f012:**
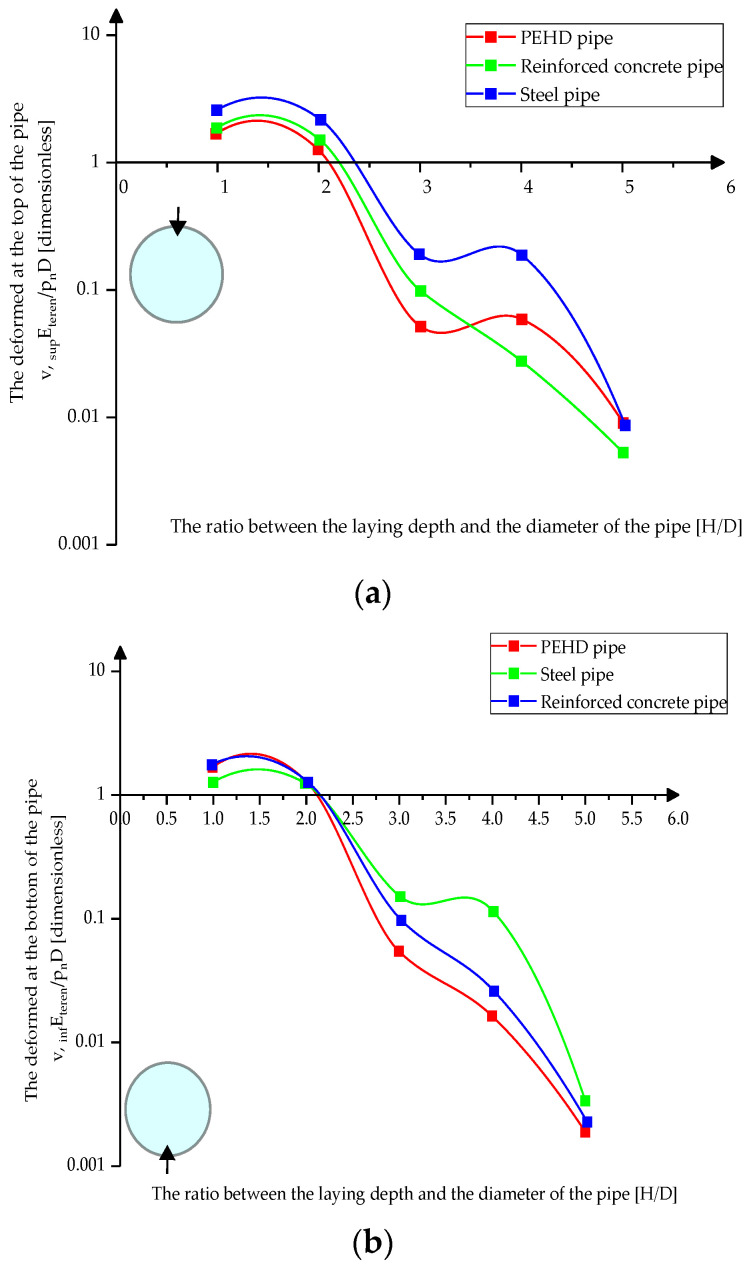
Absolute displacement variation in loose sand (**a**) at the top of the pipes (**b**) at the bottom of the pipes (**c**) in middle sections.

**Figure 13 materials-15-03385-f013:**
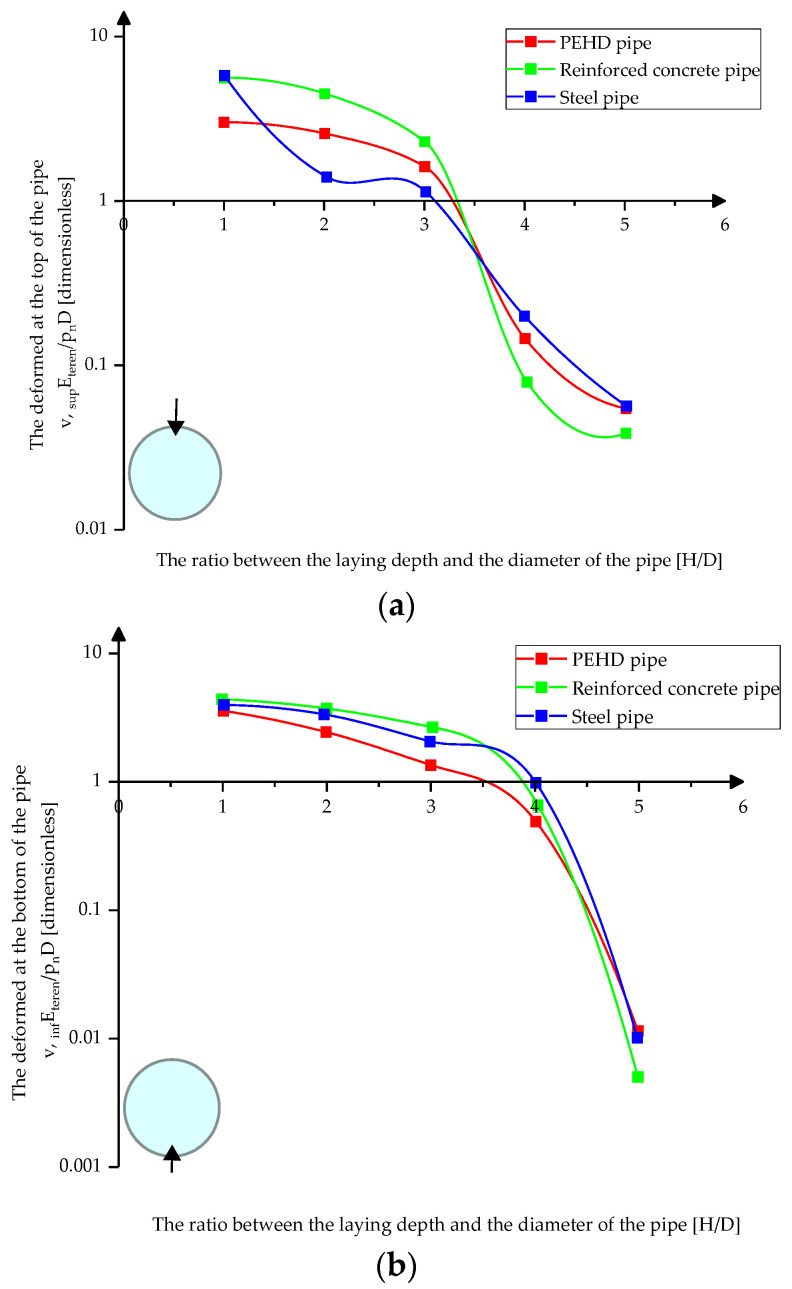
Absolute displacement variation in compacted sand (**a**) at the top of the pipes (**b**) at the bottom of the pipes (**c**) in middle sections.

**Figure 14 materials-15-03385-f014:**
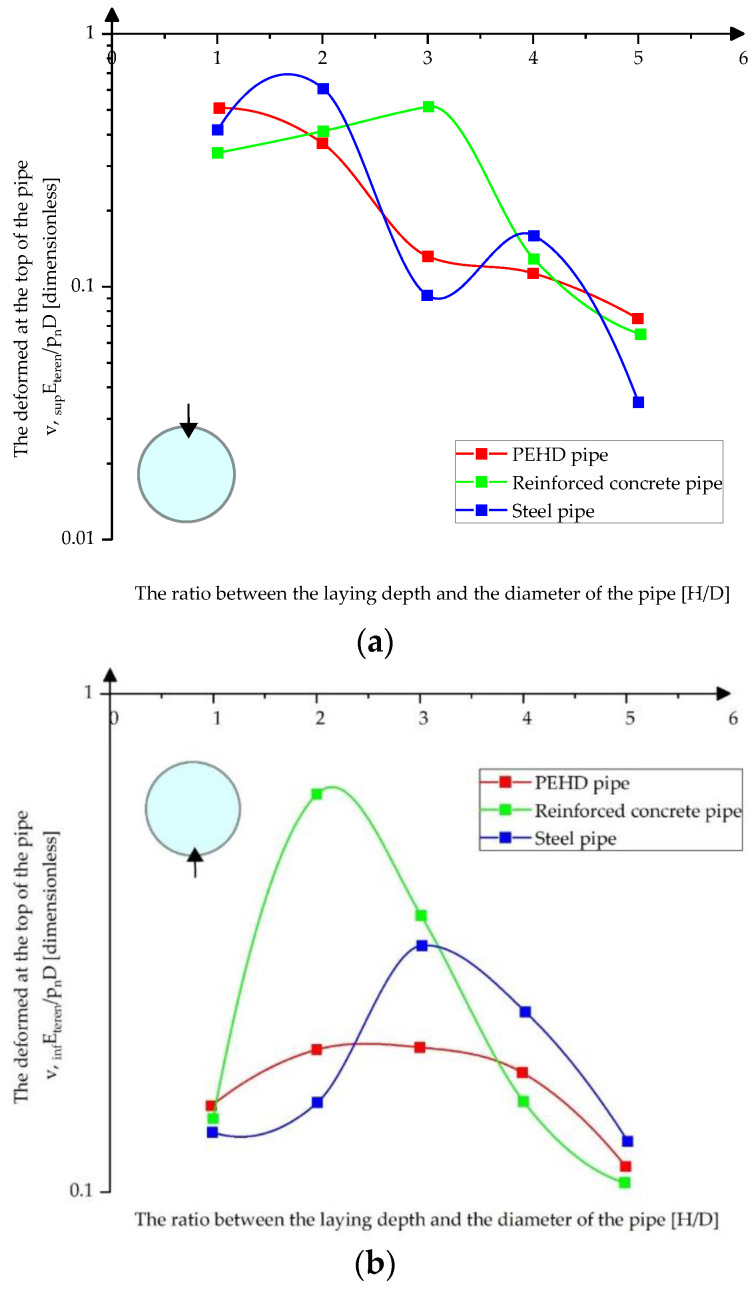
Absolute displacement variation in compacted clay (**a**) at the top of the pipes (**b**) at the bottom of the pipes.

**Table 1 materials-15-03385-t001:** Mechanical characteristics of pipes.

Material	Longitudinal Elasticity Module E (daN/cm^2^)	Poisson’s Coefficient υ	Flow Limit f_c_ (daN/cm^2^)
**Steel S235**	2.1 × 10^6^	0.30	2350
**Concrete—C.25/30**	3.5 × 10^5^	0.20	300
**High density** **polyethylene** **PEHD**	9.5 × 10^3^	0.35	255

**Table 2 materials-15-03385-t002:** Mechanical characteristics of foundation ground.

Material	Longitudinal Elasticity Module E (daN/cm^2^)	Poisson’s Coefficient υ	Flow Limit f_c_ (daN/cm^2^)
**Loose sand**	200	0.20	1650
**Compact sand**	750	0.35	2100
**Saturated clay**	1000	0.40	2300

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
