# Peer review of "Comparative Numerical Studies on the Structural Behavior of Buried Pipes Subjected to Extreme Environmental Actions"

_materials, 2022, doi:10.3390/ma15093385_

Round 1

Reviewer 1 Report

Review on “Numerical studies on the structural behavior of buried pipes subjected to extreme environmental actions”

by AncaÈ™ et al.

Manuscrip​t ID: materials-1708339

A- General Comments

The paper in hand concerns an investigation by comparative numerical studies of the structural response of three types of buried pipes, made of different materials (steel, concrete, high density polyethylene) resulting from the impact of the environment through exceptional external actions such as explosions at the surface of the land in the vicinity of the laying areas. The dynamic transient analysis of the equation of motion with the application of the explicit integration procedure was performed with the ANSYS numerical simulation program. It is claimed by the authors that the study allows designers in the field to solve difficult problems related to the quality of the laying ground of water networks – canal.

The topic of the paper is interesting, within the scope of the journal, and worthy of investigation. The originality of the work is acceptable and the study performed is adequate. However, the manuscript deserves a major revision and language proofreading (there are a lot of long sentences). I suggest that authors take into account the comments and questions below before the manuscript can be accepted for publication in Materials Journal.

B- Detailed Comments and questions

Title

The title is clear and consistent. However, I suggest to change “Numerical studies” to “Comparative Numerical Studies”.

Abstract

1- A general brief context should be added at the beginning of the abstract to know why this study is important;

2- More explicit results should be added to the end of the abstract;

3- Please explain the rationale behind testing the structural response of three types of buried pipes, made of different materials resulting from the impact of the environment through exceptional external actions?

Keywords

Keywords are ok.

1- Introduction

1- References relevant to Materials should be added, if possible;

2- The originality of the work should be more highlighted. What is new in the study with respect to recent studies in the field should be exposed briefly.

2- Theoretical considerations

1- References to the different equations presented should be provided;

2- What is the relation between these theoretical considerations and the rest of the manuscript? This should be explained briefly.

3- Materials and Methods

1- Quality of Figure 3 should be enhanced.

2- How the characteristics of Tables 1 and 2 were obtained?

4- Results

1- Please change the title of this section to “Results and discussion” and add more discussions on the interesting graphs and observations made;

2- More physical analysis are to be added to this section;

3- Why the ratio between the laying depth and the diameter of the pipe was varied from 1 to 5? What about other ranges? Are the behaviors presented conserved outside the 1-5 range of the ratio?

5- Discussion

1- I suggest to move this section to the previous one. Personally, I like to have results and discussion in the same section.

6- Conclusions

The main outputs of the manuscript in terms of applications should be highlighted.

7- References

References relevant to Materials should be added, if possible.

Author Response

Dear professor,

I have read with interest your review regarding the article ,,Numerical studies on the structural behavior of buried pipes subjected to extreme environmental actions”, that I sent for review to the ,,Materials” and I want to thank you for your appreciation and for your recommendations. I revised the manuscript according to your comments and I attach here my response to your remarks point-by-point.

Reviewer 2 Report

This paper presents numerical studies on the structural behavior of burried pipes subjected to extreme environmental actions. Numerical analysis was performed based on assumptions about various environmental conditions and situations, and the results were analyzed. The topics and research methods are interesting. However, it is difficult to properly understand this paper due to the lack of explanation in general. In order to be published in Materials, it is recommended to be modified as follows.

  1. Please highlight the originality and the purpose of the study in the introduction section. 
  2. Typos are found in several sentences, and English language proofreading is needed. Line 21, 145, 249 should be corrected. 
  3. More detailed explanation of Fig. 1 is needed. 
  4. The basis of gamma=1.4 in Equation 3? 
  5. The basis of reduction to Equation (6) should be presented.  
  6. There is no mention of Figure 3. The corresponding explanation is needed. 
  7. Rationale for the strain rate selection in Figure 4 should be presented. 
  8. The meaning of C20, 40, 60 in Figure 5 should be presented. 
  9. More analyses on the results should be added. Physical analysis on each simulation result should be presented. 
  10. In the discussion section, important outputs drawn from the analyses should be discussed. 
  11. In the conclusion section, the originality of this study should be highlighted.

Author Response

(The authors gave the same response as above.)

Round 2

Reviewer 1 Report

Thank you for taking into account my comments. The manuscript is now ready for publication in Materials Journal.

Reviewer 2 Report

The reviewer's comments are well addressed. The revised paper has enough quality for the acceptance to the Materials.